# Vaccination, Risk Factors and Outcomes of COVID-19 Infection in Patients with Psoriasis—A Single Centre Real-Life Experience from Eastern Slovakia

**DOI:** 10.3390/v14081646

**Published:** 2022-07-27

**Authors:** Janette Baloghová, Tomáš Kampe, Peter Kolarčik, Elena Hatalová

**Affiliations:** 1Department of Dermatovenerology, Faculty of Medicine, Pavol Jozef Šafárik University, 040 11 Košice, Slovakia; 2Department of Dermatovenerology, University Hospital Luis Pasteur, 040 11 Košice, Slovakia; tomaskampe@gmail.com; 3Department of Health Psychology and Research Methodology, Faculty of Medicine, Pavol Jozef Šafárik University, 040 11 Košice, Slovakia; peter.kolarcik@gmail.com; 4Olomouc University Society and Health Institute (OUSHI)—Sts Cyril and Methodius Faculty of Theology, Palacky University Olomouc, 771 47 Olomouc, Czech Republic; 5Department of Epidemiology, Faculty of Medicine, Pavol Jozef Šafárik University, 040 11 Košice, Slovakia; elena.hatalova@upjs.sk

**Keywords:** COVID-19, psoriasis, vaccination, comorbidities, biological therapy

## Abstract

Coronavirus disease (COVID-19) represents a threat for people with immune-mediated diseases. It seems that patients with psoriasis appear to have a similar SARS-CoV-2 infection rate as the general population. Our study aimed to identify factors associated with contracting COVID-19 and determining the severity of COVID-19 among psoriatic patients in a real practice setting. We conducted a cross-sectional study with 379 respondents. About one-quarter (*n* = 78; 25.8%) of the respondents who provided information on their COVID-19 (*n* = 302) status had contracted COVID-19. Most variables tested for their effect on getting COVID-19 proved to be statistically insignificant, except education, age and gender. Our study proved the protective effect of vaccination, especially the third dose, against the COVID-19 outcome. From all the potential variables, we found that non-Roma ethnicity increased the chance of being vaccinated at least once by 2.6-fold. Patients with a longer psoriasis duration had a higher chance of being vaccinated. We consider biological treatment of psoriasis safe during COVID-19. Vaccination of patients was a statistically significant protector against COVID-19. It is important to point out that only three doses of vaccine decreased with statistical significance the chance of getting the illness. Our findings should be confirmed on larger samples in further studies.

## 1. Introduction

Coronavirus disease (COVID-19) is a highly contagious infection caused by the SARS-CoV-2 virus that affects healthy people and those with immune-mediated diseases, as well. Psoriasis is a chronic, inflammatory immune-mediated skin disease independently associated with an increased risk of serious infections, but the absolute risk is low. Infections can provoke psoriasis, can occur in psoriasis, or can result from antipsoriatic treatment. However, the risk of serious infections increases with immunomodulatory treatment. Management of immunomodulatory treatment during a severe infection depends on the infection being treated, the risk of recurrent infection, any interventions that may alter that risk and whether the psoriasis is under control [1,2].

Although the US Centers for Disease Control and Prevention (CDC) does not list skin diseases in the COVID-19 list of risk factors, several diseases that are listed as risk factors in the CDC list are more common with skin diseases, e.g., type 2 diabetes and psoriasis, cardiovascular diseases and eczema, chronic kidney disease, and lupus [3].

Patients with psoriasis have been found to be more prone to thrombosis and comorbidities, which predict a worse course of COVID-19 [4]. Nevertheless, patients with psoriatic disease appear to have a similar SARS-CoV-2 infection rate as the general population, according to the results of several studies from Italy, which focused primarily on psoriatic patients receiving conventional or biological therapy [4,5]. The study of Gelfand et al. reported that the risk of developing COVID-19 in patients with autoimmune disease was twice that of controls. Furthermore, the severity of COVID-19 in psoriasis patients may also depend on major risk factors, such as smoking, male gender, older age, and comorbidities (cardiovascular disease, diabetes, obesity) [4].

Available study results portrayed patients with psoriasis as being more vulnerable to COVID-19 infection, but this association is not clear from the available research. Our study aimed to test this association on a sample of psoriatic patients in a real practice and to identify which factors (sociodemographic characteristics, comorbidities, psoriasis details and type of psoriasis therapy, vaccinations status) are associated with contracting COVID-19 and the severity of COVID-19 in psoriatic patients.

## 2. Materials and Methods

We conducted a retrospective single-centre cross-sectional study from February 2022 to April 2022 at the Department of Dermatovenerology, Faculty of Medicine, P.J. Šafárik University in Košice, Slovakia.

Eligible participants were all adult patients aged over 18 years with a diagnosis of psoriasis (mild, moderate to severe form) registered as outpatients at our department and treated for their chronic skin condition. We approached 422 of 455 registered patients, and 379 agreed to be involved in the study (response rate 89.8%). Disease-specific and general clinical data were extracted from patients’ records, and COVID-19-related data were obtained from the patients.

Participation in the study was voluntary. Patients were given a written informed consent form complying with internal clinical and ethical standards formulated by the university hospital and its Ethics Committee. The data used in this study comes from real-life clinical practice.

We collected a list of clinical data related to sociodemographics (age, gender, ethnicity, education, economic category, marital status), smoking and alcoholic status, general physical status (weight, height, waist circumference, systolic and diastolic blood pressure, BMI category), psoriasis-specific data (family history, phenotype, disease severity—Psoriasis Area Severity Index (PASI), hard-to-treat localisations, duration and treatment of psoriasis, type of biological therapy), comorbidities, information related to COVID-19 (course, symptoms, treatment), course of psoriasis during the COVID-19 pandemic and vaccination status. The course of the COVID-19 was divided into none (latent, asymptomatic), mild (loss of sense taste and smell, headache), severe (fever, upper respiratory tract infection, pneumonia), very severe (hospitalisation) and critical (death).

Fasting venous blood was collected and analysed for the biochemical marker of inflammation—CRP (C-reactive protein).

Descriptive statistics were performed initially for the demographic data, clinical features and comorbid diagnoses. We provided absolute numbers and the relative proportion in percentages for categorical variables and mean values and standard deviation values for the continuous variable. Differences between the subsamples were tested using the Mann–Whitney U-test for continuous variables and the Pearson chi-squared test for categorical variables.

The effect of clinical parameters on the likelihood of reporting being ill with COVID-19 among psoriatic patients was tested using binary logistic regression analysis. Crude effect models for single variable effects and adjusted effect models were used to determine the most loaded predictive model of COVID-19 illness from the available variables. We also used binary logistic regression analysis and similar modelling strategies to determine predictors of being vaccinated against COVID-19 among psoriatic patients.

Values of *p* < 0.05 were considered statistically significant. Data were analysed using the IBM SPSS 23.0 software package (IBM Corp. Released 2015. IBM SPSS Statistics for Windows, Version 23.0. Armonk, NY, USA: IBM Corp.).

## 3. Results

Initially, the study was composed of 379 participants; however, one-fifth of the participants (*n* = 77; 20.3%) included in the study did not provide reliable information about their COVID-19 status. Thus, the main analyses used the study sample of patients with COVID-19 status (*n* = 302; 79.7% of initial numbers), mean age 53.1 years (SD = 13.6), ranging from 18 to 86 years and consisting of 59.3% males (*n* = 179). Table 1, Table 2, Table 3, Table 4 and Table 5 provide descriptive information about sociodemographic characteristics, clinical information related to psoriasis conditions and therapy, comorbidities and information related to COVID-19.

Demographic and clinical characteristics are summarised in Table 1. The majority of patients had plaque psoriasis (90.9%; *n* = 341). Hard-to-treat body locations involved mostly the scalp (63.6%; *n* = 236) and nails (58.3%; *n* = 218); the genitals were involved only in 10.2% of patients (*n* = 38). Among 375 patients with known smoking status, 204 (54.4%) were non-smokers, 29 (7.7%) ex-smokers and 142 (37.9%) were current smokers.

Table 2 shows further characteristics of the psoriatic patients using continual variables. We can see that gender subsamples differed significantly in waist circumference, duration of psoriasis, baseline PASI value, and actual PASI score. All values with statistically significant differences were higher among men in comparison to women. When we compared patients who contracted COVID-19 with those who did not, we found only one statistically significant difference in the age of the patients: patients who contracted COVID-19 were significantly younger. In general, the patients involved in our study had suffered with psoriasis on average for 20.88 years (SD = 14.0); their mean PASI score measured at baseline (PASI at the time being first examined at our department) was 18.11, while the mean actual PASI score was 1.35 and the average value of BMI was 28.73 (Table 2).

### 3.1. COVID-19 Outcome Severity and Comorbidities

About one-quarter (*n* = 78; 25.8%) of the respondents who provided information on their COVID-19 status reported being positive for COVID-19 from the beginning of the pandemic. Patients who contracted COVID-19 provided the following symptoms: 64 patients (82.05%) had clinical signs of COVID-19, two of them died from COVID-19 (3.12%) and 14 patients (17.95%) were asymptomatic (Table 3).

The most commonly reported comorbidities were hypertension (55.3%; *n* = 208), hepatopathy (41.0%; *n* = 154), obesity with a BMI over 30.0 (36.9%; *n* = 140), psoriatic arthritis (27.4%; *n* = 103) and diabetes (19.5%; *n* = 73) without a significant relationship to COVID-19 (Table 4). Individual comorbidities and the number of comorbidities were not related to COVID-19.

The factors related to clinical characteristics and duration of psoriasis and type of psoriasis therapy did not show any statistically significant effects on the appearance of COVID-19 (Table 5).

We looked closer at the patients who died, whether they shared any similarities. We found out that both were men. One of them had been fully vaccinated with three doses of Pfizer; he was 86 years old, had a BMI of 26.22, was a non-smoker and had several comorbidities (hepatopathy, dyslipidaemia, hypertension, ischemic heart disease). He had psoriasis since the age of 29 and was being treated with apremilast at the time of death. The second man was not vaccinated; he was 66 years old, BMI 36.51, a smoker and had comorbidities (hypertension, ischemic heart disease, chronic renal disorder). He had psoriasis since the age of 26, and at the time of death was undergoing biological therapy with risankizumab. From these characteristics, we cannot conclude many similarities, except the male gender and ischemic heart disease. On the other hand, both patients were over 65 years of age, with serious comorbidities considered to be risk factors for complications during COVID-19 disease.

### 3.2. Factors Associated with Having Contracted COVID-19

Most variables involved in the study and tested for their effect on getting COVID-19 proved to be statistically insignificant (not shown), educational attainment, age and gender excluded (Table 6, Model 1). Even those variables were not all clearly associated with the dependent variable (having contracted COVID-19).

From the demographic variables, only education showed an independent crude effect with COVID-19, and this effect stayed statistically significant after further adjustment for the effect of age and gender (Table 6, Model 1). Patients with a secondary school education were more likely to get COVID-19 compared with patients with an elementary or university education, who did not differentiate with statistical significance in the likelihood of getting the illness between each other.

The effect of age and gender appeared statistically significant only in the combined effect models, where their effect was adjusted for education attainment (Table 6 Model 1, adjusted models with two predictors). This implies that higher ages decrease the likelihood of contracting COVID-19 disease. The effect of the patients’ gender, even after adjusting for education and age, did not cross the statistical significance boundary; however, it was very close, so we are eagerly inclined to interpreted it as so, because the results indicate that men have a higher chance of contracting COVID-19 in comparison to women (Table 6, Model 1, fully adjusted model with three predictors).

Taking those regression analysis effects and mutual adjustments into account (adjusted model with three predictors), we could conclude that psoriatic patients who are male, with a lower age and a secondary school education are more likely to contract the COVID-19 disease.

Similarly to Model 1, we tested the effect of vaccination on contracting the disease (Table 6, Model 2). Model 2 enriched Model 1 with vaccination status. The results confirmed that vaccination decreases the likelihood of reporting the contraction of COVID-19 disease. The protective effect was significant even after adjustment for education attainment, gender and age. It is important to point out that only three doses of vaccine decreased with statistical significance the chance of getting the disease in our sample. Having only one or two doses of vaccine did not differ with statistical significance from the effect of not being vaccinated against COVID-19.

Clinical parameters and factors related to psoriasis therapy did not show any statistically significant effects on the appearance of COVID-19.

Of the 302 patients who reported their COVID-19 status, 188 patients reported being vaccinated (62.3%): 153 (81.0%) had been vaccinated with Pfizer, 8 (4.2%) with AstraZeneca, 4 (2.1%) with Johnson & Johnson, 4 (2.1%) with Moderna, 1 (0.1%) with Sputnik and 18 (9.5%) patients had a combined vaccination (AstraZeneca and Pfizer). A total of 140 patients (74.1%) were fully vaccinated (had received three doses of vaccine), while 42 (22.2%) had received two doses and 7 (3.7%) had received only one dose of vaccine.

By comparing patients according to the type of vaccine and other variables—gender, age, comorbidities, type of systemic and biological therapy—we found no significant differences. We also checked the relationship between the type of vaccine and contracting COVID-19. The chi2 test did not show a significant difference between the vaccines but due to the total predominance of the Pfizer vaccine in our cohort (81%) and the marginal occurrence of other vaccines, it is not possible to draw a conclusion. The results of the survey are limited by the low number of respondents in our cohort. Further investigation on a larger number of patients is needed.

We found statistically significant differences between patients who contracted COVID-19 and those who did not in the vaccination rate (chi2 = 11.464; *p* < 0.01). Only 35% of patients who did not contract COVID-19 were unvaccinated, while 44.9% of those who did contract the disease were unvaccinated. Patients who did not contract COVID-19 also had a higher rate of those who had received all three doses of the vaccine (51.3%), compared to only 30.8% of patients with full vaccination in the group who did contract COVID-19 (Table 3).

### 3.3. Factors Associated with Vaccination Status

We tested variables related to vaccination against COVID-19 status among psoriatic patients in a similar way. We admit that most of the available variables were not statistically associated with vaccination status of the patients, but there were a few statistically significant associations worthy of being reported. From all the potential variables, we found that non-Roma ethnicity increased by about 3-fold the chance of being vaccinated at least once (at least one dose or more) compared with patients of Roma ethnicity. Patients with a longer duration of psoriasis also had a slightly higher chance of being vaccinated compared to patients with shorter duration of psoriasis. A lower chance of being vaccinated at least once was related to higher numbers of comorbidities and having reported the first psoriasis lesion in the scalp. Women also had a statistically significantly higher chance of being vaccinated compared to men (Table 7, Model 4).

Interesting is that these associations remained statistically significant even after mutual adjustment; however, the association of reporting a psoriasis lesion in the scalp was not tested for significance as a crude effect on vaccination status (Table 7, Model 1 vs. Model 4). Patients of non-Roma ethnicity and women, with a smaller number of comorbidities but longer duration of psoriasis and not reporting the first appearance of psoriasis lesions in their scalp have a higher chance of being vaccinated at least with one dose.

### 3.4. Biological Therapy and COVID-19 Outcome Severity

By comparing the different types of biologics used by patients for psoriasis at the time of COVID-19 positivity, we found that biological treatment of psoriasis had no effect on the COVID-19 outcome (Table 8). We would especially like to point out that biological therapy had no adverse effect on COVID-19 outcome and it did not contribute to a severe COVID-19 outcome. A total of 27.77% patients treated by anti-IL23, 24.7% of patients treated by anti-TNF and 23.88% of patients treated by anti-IL17 were positive for COVID-19. This means that biological therapy subgroups did not differ significantly in the prevalence of positive COVID-19 patients. Of the 260 patients treated with biologics, 67 had COVID-19 (25.77%), but only 5 patients were hospitalised and only 1 patient died due to COVID-19.

The most common course of COVID-19 was mild (66.7%; *n* = 52) and the most common symptoms were respiratory (71.8%; *n* = 56). Psoriasis worsened in 8 patients (10.3%), and in 6 patients psoriatic therapy was interrupted. Psoriasis was treated during COVID-19 disease in the majority of the patients (79.5%; *n* = 62) with biological therapy (Table 8).

## 4. Discussion

Although patients with psoriasis are more prone to infections, published data show that COVID-19 affects fewer patients with psoriasis than was expected [5,6,7]. A cross-sectional study of Yiu et al. reported that psoriasis (regardless of treatment) was not associated with an increase in the risk of testing positive for COVID-19 [8]. Gisondi et al. reported that there is no early signal of increased hospitalisation or death from COVID-19 for psoriatic patients [5]. In our cohort, only about one-quarter of psoriatic patients had been infected.

Predictive factors found in the general population may be decisive. Factors including age (≥65 years), gender (male), pre-existing comorbidities (linking cardiovascular functions, hypertension, thrombosis, diabetes, chronic liver disease), immune response, laboratory markers and indicators of organ dysfunction may predict worse outcomes of COVID-19 [9,10,11,12].

In our cohort, we did not notice a higher incidence of COVID-19 in patients with comorbidities (cardiovascular disease, ischemic heart disease, diabetes mellitus, hypertension, hepatopathy and chronic kidney disease), even in connection with obesity. From the 78 patients with a positive test for COVID-19, seven had a very severe course and two died.

Based on our findings, we could say that psoriatic patients who are male, with a lower age and secondary school education have a higher likelihood of getting COVID-19. We assume that lower-aged male patients in our cohort with secondary school education having a shorter coexistence with the psoriatic disease also have lower adherence to treatment and disease management, as well. Surely, the virus mutation and the time factor also have an effect on the COVID-19 outcome.

At the outset of the COVID-19 pandemic, much consideration was given to whether it is necessary to discontinue immunosuppressive and especially biological treatment for psoriasis, whether biological treatment could affect the onset or worsening of COVID-19. In an international case series of patients with moderate-to-severe psoriasis published by Mahil et al., biological therapy was associated with a lower risk of COVID-19-related hospitalisation than other systemic therapies [12]. Patients with moderate-to-severe psoriasis on a biological agent have a similar or perhaps even a lower incidence of COVID-19 compared to the general public [13].

The use of the biologics for psoriasis in our cohort was not associated with increased risk or a worse COVID-19 outcome. From the 265 patients treated with biologics, 61 patients (23.02%) had COVID-19, 7 of them were hospitalised and only 1 patient with psoriasis being treated with biological therapy died due to COVID-19. Overall, patients on biologics continued the therapy. Six patients discontinued biological treatment at the time of COVID-19 (due to symptoms) without worsening their symptoms of psoriasis.

Plasma concentrations of proinflammatory molecules in patients with SARS-CoV-2 are higher than those in healthy controls, including IL-12, IL-17 and TNF-α, which play an important role in the pathogenesis of psoriasis [14]. TNF-α is secreted by different types of immune cells, including monocytes, lymphocytes or fibroblasts. Increased plasma levels of TNF-α in COVID-19 are associated with disease severity and inversely correlate with the reduction in T lymphocytes. It has been reported that the prevalence of severe cases of COVID-19 was lower in patients with anti-TNF-α therapy compared to patients treated with steroids [15]. IL-17 plays a key role in adaptive immunity and inflammatory responses in the body during infection as well as during severe COVID-19 disease. It is important to emphasise the importance of Th-17-type cytokine storm in the pathogenesis of COVID-19, IL-17 is produced by Th17 cells which are increased leading to increased production of IL-17 and IL-22 cytokines. Elevated IL-17 levels in patients with SARS-CoV-2 have been associated with the viral load and disease severity. The combination of IL-17 with TNF-α induces the expression of pro-coagulation factors, which promotes thrombosis and inhibits the endothelial anticoagulatory pathway. IL-17 seems to increase the replication of some viruses and leads to viral persistence [16,17]. IL-12 is mainly produced by dendritic cells, macrophages and B lymphocytes, and acts as an immunoregulatory factor that can promote the proliferation of Th1 and Th17 cells. IL-12 plays an important role in cytokine storm by augmenting the activation of various immune cells. IL-12 has the ability to establish links between innate and adaptive immune responses and acts on its receptor (IL-12R). The increased serum concentration of this interleukin has been shown in patients with high COVID-19 infection [18,19].

It has been suggested that inhibitors that can block IL-12, IL-17 and TNF-α can be used to treat COVID-19 infections [6,20]. Biological treatment targeted against TNF-α and IL-17 could play an important protective role against COVID-19 in psoriatic patients.

However, which biological agent has the most pronounced effect on COVID-19 has still not been determined [6]. In our cohort, no concrete group of biological agents had a protective effect against COVID-19 outcome.

Patients with psoriasis are more susceptible to infections due to the use of immunomodulating and immunosuppressive treatments. Therefore, vaccination is recommended as a prevention of specific infections (although it can lead to the initiation or exacerbation of psoriasis). Live-attenuated vaccines, formed by weakened natural pathogens, induce a permanent and rapid humoral response. By mimicking natural infection, vaccination with such a vaccine is usually associated with the symptoms of an active viral infection. Therefore, live-attenuated vaccines are contraindicated in immunocompromised patients. Inactivated vaccines are considered safer, containing the killed pathogen, purified pathogen antigens or inactivated toxins. Although it is usually necessary to repeat vaccination with this vaccine (boosters), they are advantageous as they do not contain an active pathogen [21].

Regarding the vaccination against COVID-19, all the vaccines against COVID-19 approved so far are inactivated. Nonlive vaccines are divided into virus-based or protein-based vaccines. A new subgroup of vaccines is based on nucleic acids (mRNA/DNA). Their effective component is genetic information. Antigen production takes place in the muscle cells and immune cells of the vaccinated person. The presentation of the antigen to the immune system thus resembles a natural infection [22]. At present, multiple coronavirus vaccines have been developed, including the mRNA vaccines BNT162b2 (Pfizer-BioNTech/Comirnaty), mRNA-1273 (Moderna), and an adenoviral vector vaccine Ad26.COV2.S (Johnson & Johnson/Janssen), the AZD1222 (Oxford-AstraZeneca/Covidshield), the inactivated vaccine CoronaVac (Sinovac Biotech), WIBP/BBIBP-CorV COVID-19 vaccine (Sinopharm) and the adenoviral vector vaccine Gam-COVID-Vac (Sputnik V) [23]. Currently, there are over 100 candidate SARS-CoV-2 vaccines under development. Most candidate vaccines target surface membrane S protein, which is involved in receptor binding, membrane fusion and entry into host cells [24]. Vaccination of patients in our cohort was a statistically significant protector against COVID-19. It is important to point out that only three doses of vaccine decreased with statistical significance the chance of getting the illness. Patients with a longer duration of psoriasis had a higher chance of being vaccinated. Only 1 patient out of 189 vaccinated patients showed worsened psoriasis after vaccination, but this patient missed two doses of secukinumab injections.

Younger patients who overcame COVID-19 in our cohort were not vaccinated. The reason why they refused vaccinations may be growing scepticism and vaccine hesitancy influenced by social media and misinformation. The situation with elderly patients should be associated with a better perception of their personal health; they may trust their doctor more and get advice, or they may be more worried about themselves, i.e., not getting COVID-19; thus, they prefer to be vaccinated.

Our findings are consistent with the case study of Damiani et al. that claims RNA-based vaccines to be safe and effective for psoriatic patients undergoing immunosuppresive therapy and not worsening psoriasis [25].

Our analysis showed one strange effect that probably needs further examination. We found that patients who report the primary occurrence of psoriasis in the scalp was associated with lower chance of being vaccinated against COVID-19. The association appeared only in an adjusted regression model, but it might indicate some sort of specificity of the patients whose first symptoms manifested in the scalp as a specific and psycho-socially sensitive location. Patients become more confined from society; thus, they may lose confidence in treatment and possibly in vaccination as well.

We consider the fact that the study provides real life insight into the effect of the characteristics of psoriatic patients related to the main diagnosis, treatment and comorbidities on the likelihood to contract COVID-19 and the effect of psoriatic therapy, especially biological therapy, on the COVID-19 outcome and effect of vaccination to COVID-19 infection and its outcome as a strength of our study. Our finding provides a picture of the prevalence of the disease after fourth wave of the COVID-19 pandemic (the omicron variant).

Limitations of the study include the number of participants—this limited some analyses where the subsamples were too small to yield statistically significant associations; data coming from only one centre (but from the endpoint hospital for a large region); and no control sample from the general population (the inability to compare at least the prevalence of COVID-19 in the general population with our sample; we cannot say whether the prevalence among psoriatic patients is higher or lower than in the general population).

## 5. Conclusions

We did not find any major risk for contracting COVID-19 among psoriatic patients related to their psoriatic symptoms, treatment or comorbidities. Major risk factors were younger age, male gender and secondary school education attainment. We can consider biological treatment of psoriasis as safe or providing no extra health risks during COVID-19; thus, not requiring interruption to prevent the risk of infection. Our study showed a protective effect of vaccination, especially the booster, the third dose, against the COVID-19 outcome.

## Figures and Tables

**Table 1 viruses-14-01646-t001:** Sociodemographic characteristics and psoriatic categories (numbers and relative prevalence in %) presented for the total sample and stratified according to gender and COVID-19 contraction status, differences tested with Pearson’s chi-squared test.

	Total Sample (N = 379)	Gender (N = 379)		Gender Differences Chi2 Value *p*-Value	COVID-19 Status (N = 302)		COVID-19 Status Differences Chi2 Value *p*-Value
		Males	Females		Not Contracted	Contracted	
	N (%)	N (%)	N (%)		N (%)	N (%)	
**Gender**							
male	212 (56.1%)				128 (57.1%)	51 (65.4%)	1.628
female	166 (43.9%)				96 (42.9%)	27 (34.6%)	*0.202*
**Ethnicity**							
Non-Roma	330 (87.5%)	186 (88.2%)	144 (87.3%)	0.067	197 (88.3%)	71 (92.2%)	0.898
Roma	47 (12.5%)	25 (11.8%)	21 (12.7%)	*0.796*	26 (11.7%)	6 (7.8%)	*0.343*
**Education**							
elementary	111 (29.7%)	64 (30.5%)	46 (28.2%)	6.062	60 (27.1%)	13 (17.1%)	6.541
secondary	187 (50%)	95 (45.2%)	92 (56.4%)	* **0.048** *	111 (50.2%)	51 (67.1%)	* **0.038** *
university	76 (20.3%)	51 (24.3%)	25 (15.3%)		50 (22.6%)	12 (15.8%)	
**Economic category**							
employed	226 (60.3%)	142 (67.3%)	84 (51.5%)	14.921	130 (58.6%)	53 (69.7%)	7.883
unemployed	34 (9.1%)	16 (7.6%)	17 (10.4%)	* **0.011** *	18 (8.1%)	7 (9.2%)	*0.163*
student	14 (3.7%)	3 (1.4%)	11 (6.7%)		9 (4.1%)	1 (1.3%)	
retired	83 (22.1%)	42 (19.9%)	41 (25.2%)		56 (25.2%)	10 (13.2%)	
medically retired	18 (4.8%)	8 (3.8%)	10 (6.1%)		9 (4.1%)	5 (6.6%)	
**Marital status**							
single	68 (18.3%)	41 (19.5%)	26 (16%)	4.212	33 (14.9%)	18 (23.7%)	3.749
married	272 (72.9%)	154 (73.3%)	118 (72.8%)	*0.378*	164 (74.2%)	51 (67.1%)	*0.441*
widowed	19 (5.1%)	7 (3.3%)	12 (7.4%)		13 (5.9%)	4 (5.3%)	
divorced	14 (3.8%)	8 (3.8%)	6 (3.7%)		11 (5%)	3 (3.9%)	
**BMI category**							
<18.5 underweight	7 (1.8%)	0 (0%)	7 (4.2%)	**16.605**	3 (1.3%)	2 (2.6%)	4.520
18.5–24.9 normal weight	88 (23.2%)	42 (19.8%)	7 (4.2%)	* **0.011** *	56 (25%)	12 (15.4%)	*0.607*
25.0–29.9 overweight	144 (38%)	92 (43.4%)	45 (27.1%)		83 (37.1%)	33 (42.3%)	
30–34.9 obese (class I)	88 (23.2%)	47 (22.2%)	52 (31.3%)		51 (22.8%)	19 (24.4%)	
35–39.9 obese (class II)	30 (7.9%)	20 (9.4%)	41 (24.7%)		18 (8%)	7 (9%)	
over 40.0 obese (class III)	22 (5.8%)	11 (5.2%)	10 (6%)		13 (5.8%)	5 (6.4%)	
**Smoking status**							
non-smoker (never)	204 (54.4%)	101 (48.1%)	103 (62.8%)	**12.971**	127 (57.5%)	43 (55.8%)	0.528
current smoker	142 (37.9%)	85 (40.5%)	56 (34.1%)	* **0.002** *	77 (34.8%)	27 (35.1%)	*0.913*
ex-smoker	29 (7.7%)	24 (11.4%)	5 (3%)		17 (7.7%)	7 (9.1%)	
**Alcohol status**							
abstainer	75 (20%)	25 (11.9%)	50 (30.5%)	**24.983**	41 (18.6%)	19 (24.7%)	2.161
occasional drinker	283 (75.5%)	170 (81%)	112 (68.3%)	* **0.000** *	171 (77.4%)	54 (70.1%)	*0.540*
regular drinker	16 (4.3%)	14 (6.7%)	2 (1.2%)		8 (3.6%)	4 (5.2%)	
quit drinking	1 (0.3%)	1 (0.5%)	0 (0%)		1 (0.5%)	0 (0%)	
**PsO phenotype**							
plaque	341 (90.9%)	196 (92.9%)	144 (88.3%)	9.551	213 (95.9%)	74 (97.4%)	4.961
palmo-plantar	22 (5.9%)	9 (4.3%)	13 (8%)	*0.145*	8 (3.6%)	1 (1.3%)	*0.291*
inverse	1 (0.3%)	0 (0%)	1 (0.6%)		1 (0.5%)	0 (0%)	
guttate	5 (1.3%)	3 (1.4%)	2 (1.2%)		0 (0%)	1 (1.3%)	
pustular	6 (0.5%)	3 (1.4%)	3 (1.8%)		0 (0%)	0 (0%)	
**PsO**—**nails**							
uninvolved	156 (41.7%)	87 (41.4%)	69 (42.3%)	0.031	81 (36.7%)	26 (34.2%)	0.146
involved	218 (58.3%)	123 (58.6%)	94 (57.7%)	*0.861*	140 (63.3%)	50 (65.8%)	*0.702*
**PsO**—**scalp**							
uninvolved	136 (36.4%)	71 (33.8%)	65 (39.9%)	1.458	73 (33%)	22 (28.9%)	0.434
involved	238 (63.6%)	139 (66.2%)	98 (60.1%)	*0.227*	148 (67%)	54 (71.1%)	*0.510*
**PsO**—**genitalia**							
uninvolved	336 (89.8%)	189 (90%)	147 (90.2%)	0.003	199 (90%)	68 (89.5%)	0.020
involved	38 (10.2%)	21 (10%)	16 (9.8%)	*0.953*	22 (10%)	8 (10.5%)	*0.887*
**PASI category**							
PASI < 10 mild PsO	49 (13.5%)	13 (6.4%)	36 (22.8%)	**20.323**	14 (6.3%)	5 (6.5%)	0.002
PASI > 10 moderate to severe PsO	313 (86.5%)	190 (93.6%)	122 (77.2%)	* **0.000** *	207 (93.7%)	72 (93.5%)	*0.961*
**Family history**—**PsO**							
negative	232 (62.5%)	129 (62%)	102 (63%)	1.571	131 (59.8%)	50 (65.8%)	1.402
positive	139 (37.5%)	79 (38%)	60 (37%)	*0.456*	88 (40.2%)	26 (34.2%)	*0.496*

Abbreviations: BMI—Body Mass Index; PASI—Psoriasis Area Severity Index; PsO—psoriasis; Note: Statistically significant differences are highlighted in bold.

**Table 2 viruses-14-01646-t002:** Means and standard deviation of the continual variables presented for the total sample, stratified according to gender and COVID-19 contraction; differences tested with Mann–Whitney U-test.

	Total Sample (*n* = 379)		Gender (*n* = 379)				Gender Differences		COVID-19 Status (*n* = 302)				COVID-19 Status Differences	
			Men		Women				Not Contracted		Contracted			
	M	*SD*	M	*SD*	M	*SD*	U Value	*p*-Value	M	*SD*	M	*SD*	U Value	*p*-Value
**Age**	52.24	*13.99*	52.03	*12.87*	52.61	*15.33*	16,237.5	*0.395*	53.95	*13.99*	50.54	*12.52*	6933.5	* **0.023** *
**Waist circumference**	98.74	*17.61*	103.10	*16.24*	93.53	*17.88*	6767.5	* **0.000** *	99.97	*17.77*	102.71	*16.61*	3620.5	*0.276*
**BMI value**	28.73	*5.68*	29.10	*5.17*	28.30	*6.26*	16,179.0	*0.179*	28.77	*5.74*	28.94	*5.63*	8418.0	*0.632*
**PsO duration**	20.88	*13.96*	22.14	*13.98*	19.26	*13.84*	14,836.0	* **0.027** *	22.92	*14.66*	21.45	*12.84*	8097.0	*0.641*
**PASI baseline score**	18.11	*9.04*	19.69	*8.70*	15.98	*9.08*	13,410.0	* **0.000** *	18.95	*7.98*	21.61	*9.73*	8495.0	*0.961*
**PASI actual score**	1.35	*3.82*	1.08	*4.10*	1.73	*3.39*	9059.0	* **0.003** *	1.36	*4.20*	1.11	*2.22*	7361.0	*0.809*
**Number of comorbidities**	2.99	*1.97*	3.09	*1.90*	2.88	*2.06*	15,536.5	*0.186*	3.14	*2.01*	3.17	*1.85*	8114.5	*0.830*
**CRP**	5.39	*7.89*	5.48	*9.07*	5.30	*6.11*	15,960.5	*0.225*	5.22	*7.45*	4.89	*6.44*	8199.0	*0.486*

Abbreviations: BMI—body mass index; PASI—Psoriasis Area Severity Index; PsO, psoriasis; CRP—C-reactive protein; M—mean; SD—standard deviation. Note: statistically significant differences are highlighted in bold.

**Table 3 viruses-14-01646-t003:** Vaccination status, COVID-19 symptoms and treatment, impact on psoriasis and treatment variables (numbers and relative prevalence in %) presented for the total sample and stratified according to gender and COVID-19 contraction status, differences tested with Pearson’s chi-squared test.

	Total Sample	Gender		Gender Differences Chi2 Value *p*-Value	COVID-19		COVID-19 Differences Chi2 Value *p*-Value
		Males	Females		Not Contracted	Contracted	
	N (%)	N (%)	N (%)		N (%)	N (%)	
**Vaccination against COVID-19**							
none	118 (38.4%)	59 (32.4%)	59 (47.2%)	**9.280**	79 (35.3%)	35 (44.9%)	**11.464**
1st dose	7 (2.3%)	6 (3.3%)	1 (0.8%)	* **0.026** *	5 (2.2%)	2 (2.6%)	* **0.009** *
2nd dose	42 (13.7%)	24 (13.2%)	18 (14.4%)		25 (11.2%)	17 (21.8%)	
3rd dose (booster)	140 (45.6%)	93 (51.1%)	47 (37.6%)		115 (51.3%)	24 (30.8%)	
**COVID-19 course of illness**							
none	14 (17.9%)	10 (19.6%)	4 (14.8%)	2.161		14 (17.9%)	
mild	52 (66.7%)	32 (62.7%)	20 (74.1%)	*0.706*		52 (66.7%)	
severe	5 (6.4%)	3 (5.9%)	2 (7.4%)			5 (6.4%)	
very severe	5 (6.4%)	4 (7.8%)	1 (3.7%)			5 (6.4%)	
critical (death)	2 (2.6%)	2 (3.9%)	0 (0%)			2 (2.6%)	
**COVID-19 symptoms**							
none	15 (19.2%)	11 (21.6%)	4 (14.8%)	2.710		15 (19.2%)	
respiratory	56 (71.8%)	34 (66.7%)	22 (81.5%)	*0.607*		56 (71.8%)	
gastrointestinal	4 (5.1%)	3 (5.9%)	1 (3.7%)			4 (5.1%)	
neurological	1 (1.3%)	1 (2%)	0 (0%)			1 (1.3%)	
others	2 (2.6%)	2 (3.9%)	0 (0%)			2 (2.6%)	
**COVID-19 therapy**							
none	23 (29.5%)	17 (33.3%)	6 (22.2%)	8.478		23 (29.5%)	
symptomatic vitamins. antiflogistics	29 (37.2%)	14 (27.5%)	15 (55.6%)	*0.132*		29 (37.2%)	
ATB, Isoprinosine	17 (21.8%)	13 (25.5%)	4 (14.8%)			17 (21.8%)	
corticosteroids	4 (5.1%)	2 (3.9%)	2 (7.4%)			4 (5.1%)	
remdesivir	3 (3.8%)	3 (5.9%)	0 (0%)			3 (3.8%)	
ALV	2 (2.6%)	2 (3.9%)	0 (0%)			2 (2.6%)	
**PsO during COVID-19**							
clear skin	53 (67.9%)	35 (68.6%)	18 (66.7%)	1.916		53 (67.9%)	
moderate (BSA < 10%)	21 (26.9%)	14 (27.5%)	7 (25.9%)	*0.590*		21 (26.9%)	
severe (BSA > 10%)	3 (3.8%)	2 (3.9%)	2 (7.4%)			4 (5.1%)	
**PsO reaction to COVID-19**							
no change	70 (89.7%)	45 (88.2%)	25 (92.6%)	0.364		70 (89.7%)	
worsened status	8 (10.3%)	6 (11.8%)	2 (7.4%)	*0.546*		8 (10.3%)	
**PsO therapy during COVID-19**							
topical	6 (7.7%)	3 (5.9%)	3 (11.1%)	10.123		6 (7.7%)	
methotrexate	1 (1.3%)	1 (2%)	0 (0%)	*0.519*		1 (1.3%)	
cyclosporine A	3 (3.8%)	1 (2%)	2 (7.4%)			3 (3.8%)	
apremilast	6 (7.7%)	4 (7.8%)	2 (7.4%)			6 (7.7%)	
adalimumab	15 (19.2%)	9 (17.6%)	6 (22.2%)			15 (19.2%)	
etanercept	4 (5.1%)	3 (5.9%)	1 (3.7%)			4 (5.1%)	
ustekinumab	1 (1.3%)	1 (2%)	0 (0%)			1 (1.3%)	
secukinumab	6 (7.7%)	2 (3.9%)	4 (14.8%)			6 (7.7%)	
ixekizumab	8 (10.3%)	5 (9.8%)	3 (11.1%)			8 (10.3%)	
brodalumab	1 (1.3%)	1 (2%)	0 (0%)			1 (1.3%)	
risankizumab	23 (29.5%)	19 (37.3%)	4 (14.8%)			23 (29.5%)	
guselkumab	4 (5.1%)	2 (3.9%)	2 (7.4%)			4 (5.1%)	
**COVID-19 illness**							
not contracted	224 (74.2%)	128 (71.5%)	96 (78%)	1.628			
contracted	78 (25.8%)	51 (28.5%)	27 (22%)	*0.202*			
**Worsened PsO after vaccination**							
no	188 (99.5%)	122 (99.2%)	66 (100%)	0.539	144 (99.3%)	43 (100%)	
yes	1 (0.5%)	1 (0.8%)	0 (0%)	*0.463*	1 (0.7%)	0 (0%)	
**PsO treatment discontinuation**							
no	67 (91.8%)	45 (93.8%)	22 (88.0%)	0.720		67 (91.8)	
yes	6 (8.2%)	3 (6.3%)	3 (12.0%)	*0.369*		6 (8.2)	
**Type of biological therapy**							
antiTNF	82 (30.9%)	52 (31.3%)	30 (30.3%)	3.343	55 (29.1%)	21 (31.3%)	
anti-IL17	67 (25.3%)	36 (21.7%)	31 (31.3%)	*0.188*	51 (27%)	16 (23.9%)	
anti-IL23	116 (43.8%)	78 (47%)	38 (38.4%)		83 (43.9%)	30 (44.8%)	

Abbreviations: ATB—antibiotics; ALV—Artificial Lung Ventilation; PsO—psoriasis; TNF—Tumour necrosis factor; Note: Statistically significant differences are highlighted in bold.

**Table 4 viruses-14-01646-t004:** Summarisation of major comorbidities of psoriatic patients (numbers and relative prevalence in %) presented for the total sample and stratified according to gender and COVID-19 contraction status, differences tested with Pearson’s chi-squared test.

	Total Sample	Gender		Gender Differences Chi^2^ Value *p* Value	COVID-19 Status		COVID-19 Status Differences Chi^2^ Value *p* Value
		Males	Females		Not Contracted	Contracted	
	N (%)	N (%)	N (%)		N (%)	N (%)	
**Hypertension**							
no	168 (44.7%)	90 (42.9%)	77 (46.7%)	0.543	94 (42.5%)	30 (38.5%)	0.394
yes	208 (55.3%)	120 (57.1%)	88 (53.3%)	*0.461*	127 (57.5%)	48 (61.5%)	*0.530*
**Diabetes mellitus**							
no	302 (84.8%)	163 (81.1%)	138 (89.6%)	**4.903**	173 (82%)	64 (88.9%)	1.877
yes	54 (15.2%)	38 (18.9%)	16 (10.4%)	** *0.027* **	38 (18%)	8 (11.1%)	0.171
**PsA**							
no	273 (72.6%)	157 (74.1%)	116 (70.7%)	0.514	151 (68.0%)	55 (71.4%)	0.310
yes	103 (27.4%)	55 (25.9%)	48 (29.3%)	*0.473*	71 (32%)	22 (28.6%)	*0.577*
**Dyslipidemia**							
no	117 (31.2%)	60 (28.4%)	57 (34.8%)	1.717	69 (31.2%)	24 (31.2%)	0.000
yes	258 (68.8%)	151 (71.6%)	107 (65.2%)	*0.190*	152 (68.8%)	53 (68.8%)	*0.993*
**Hepatopathy**							
no	222 (59%)	111 (52.4%)	111 (67.7%)	**18.577**	133 (59.9%)	43 (55.8%)	4.576
yes	154 (41%)	101 (47.6%)	53 (32.3%)	** *0.005* **	89 (40.1%)	34 (44.2%)	*0.599*
**Ischemic heart disease**							
no	344 (91.5%)	190 (90%)	153 (93.3%)	1.245	199 (89.6%)	70 (90.9%)	0.102
yes	32 (8.5%)	21 (10%)	11 (6.7%)	*0.264*	23 (10.4%)	7 (9.1%)	*0.749*
**Myocardial infarction**							
no	364 (96.8%)	202 (95.7%)	161 (98.2%)	1.768	212 (95.5%)	75 (97.4%)	0.540
yes	12 (3.2%)	9 (4.3%)	3 (1.8%)	*0.184*	10 (4.5%)	2 (2.6%)	*0.463*
**Stroke**							
no	369 (98.1%)	208 (98.6%)	160 (97.6%)	0.521	216 (97.3%)	77 (100%)	2.124
yes	7 (1.9%)	3 (1.4%)	4 (2.4%)	*0.470*	6 (2.7%)	0 (0%)	*0.145*
**Heart failure**							
no	366 (97.3%)	202 (95.7%)	163 (99.4%)	**4.751**	216 (97.3%)	74 (96.1%)	0.279
yes	10 (2.7%)	9 (4.3%)	1 (0.6%)	** *0.029* **	6 (2.7%)	3 (3.9%)	*0.597*
**Peripheral artery disease**							
no	359 (95.7%)	198 (93.8%)	160 (98.2%)	**4.192**	212 (95.9%)	73 (94.8%)	0.172
yes	16 (4.3%)	13 (6.2%)	3 (1.8%)	** *0.041* **	9 (4.1%)	4 (5.2%)	*0.678*
**Cancer**							
negative	360 (96%)	202 (96.2%)	157 (95.7%)	11.829	215 (96.8%)	70 (92.1%)	6.897
NMSC	6 (1.6%)	5 (2.4%)	1 (0.6%)	*0.066*	3 (1.4%)	3 (3.9%)	*0.330*
MM	1 (0.3%)	1 (0.5%)	0 (0%)		1 (0.5%)	0 (0%)	
breast	2 (0.5%)	0 (0%)	2 (1.2%)		1 (0.5%)	0 (0%)	
haematological	3 (0.8%)	0 (0%)	3 (1.8%)		1 (0.5%)	1 (1.3%)	
digestive tract	2 (0.5%)	2 (1%)	0 (0%)		1 (0.5%)	1 (1.3%)	
other	1 (0.3%)	0 (0%)	1 (0.6%)		0 (0%)	1 (1.3%)	
**GIT diseases**							
negative	344 (91.5%)	195 (92.4%)	148 (90.2%)	0.874	201 (90.5%)	70 (90.9%)	3.483
Crohn disease	9 (2.4%)	5 (2.4%)	4 (2.4%)	*0.972*	5 (2.3%)	2 (2.6%)	*0.626*
Ulcerative colitis	2 (0.5%)	1 (0.5%)	1 (0.6%)		1 (0.5%)	1 (1.3%)	
celiac disease	2 (0.5%)	1 (0.5%)	1 (0.6%)		0 (0%)	1 (1.3%)	
gastritis	5 (1.3%)	2 (0.9%)	3 (1.8%)		4 (1.8%)	1 (1.3%)	
other	14 (3.7%)	7 (3.3%)	7 (4.3%)		11 (5%)	3 (3.9%)	
**Kidney disease**							
no	362 (96.5%)	206 (97.6%)	155 (95.1%)	1.766	212 (95.5%)	74 (97.4%)	0.514
yes	13 (3.5%)	5 (2.4%)	8 (4.9%)	*0.184*	10 (4.5%)	2 (2.6%)	*0.473*

Abbreviations: PsA—psoriatic arthritis; NMSC—non-melanoma skin cancer; MM—malignant melanoma; GIT—gastrointestinal tract. Note: statistically significant differences are highlighted in bold.

**Table 5 viruses-14-01646-t005:** Treatment of psoriasis and biological therapy categories (numbers and relative prevalence in %) presented for the total sample and stratified according to gender and COVID-19 contraction status, differences tested with Pearson’s chi-squared test.

	Total Sample	Gender		Gender Differences Chi2 Value *p*-Value	COVID-19 Status		COVID-19 Status Differences Chi2 Value *p*-Value
Treatment of Psoriasis		Males	Females		Not Contracted	Contracted	
	N (%)	N (%)	N (%)		N (%)	N (%)	
**Type of the treatment**							
none	12 (3.2%)	5 (2.4%)	7 (4.2%)	**23.817**	0 (0%)	0 (0%)	8.560
topical	33 (8.8%)	9 (4.3%)	24 (14.5%)	** *0.001* **	3 (1.4%)	0 (0%)	*0.200*
phototherapy	1 (0.3%)	1 (0.5%)	0 (0%)		2 (0.9%)	4 (5.3%)	
acitretine	13 (3.5%)	7 (3.4%)	6 (3.6%)		1 (0.5%)	0 (0%)	
cyclosporine A	8 (2.1%)	4 (1.9%)	4 (2.4%)		3 (1.4%)	1 (1.3%)	
methotrexate	13 (3.5%)	3 (1.4%)	9 (5.5%)		1 (0.5%)	0 (0%)	
apremilast	33 (8.8%)	15 (7.2%)	18 (10.9%)		26 (11.7%)	5 (6.6%)	
biological therapy	260 (69.7%)	163 (78.7%)	97 (58.8%)		186 (83.8%)	66 (86.8%)	
**Current biological therapy**							
adalimumab	66 (24.8%)	44 (26.5%)	22 (22%)	**22.670**	43 (22.6%)	17 (25.4%)	5.978
etanercept	17 (6.4%)	8 (4.8%)	9 (9%)	** *0.012* **	13 (6.8%)	4 (6%)	*0.742*
ustekinumab	6 (2.3%)	4 (2.4%)	2 (2%)		5 (2.6%)	1 (1.5%)	
secukinumab	33 (12.4%)	14 (8.4%)	19 (19%)		28 (14.7%)	5 (7.5%)	
ixekizumab	24 (9%)	19 (11.4%)	5 (5%)		16 (8.4%)	8 (11.9%)	
brodalumab	10 (3.8%)	3 (1.8%)	7 (7%)		7 (3.7%)	3 (4.5%)	
risankizumab	84 (31.6%)	59 (35.5%)	25 (25%)		57 (30%)	25 (37.3%)	
guselkumab	24 (9%)	15 (9%)	9 (9%)		20 (10.5%)	4 (6%)	
certolizumab	1 (0.4%)	0 (0%)	1 (1%)		1 (0.5%)	0 (0%)	
infliximab	1 (0.4%)	0 (0%)	1 (1%)		0 (0%)	0 (0%)	

Note: statistically significant differences are highlighted in bold.

**Table 6 viruses-14-01646-t006:** Logistic regression models testing the crude and adjusted effects of three sociodemographic variables on contracting COVID-19 (Model 1) and testing the crude and adjusted effects of vaccination status with the sociodemographic variables on contracting COVID-19 (Model 2).

	Crude Effect	Adjusted Models				
		2 Predictors	2 Predictors	2 Predictors	3 Predictors	4 Predictors
	OR (95% C.I.)	OR (95% C.I.)	OR (95% C.I.)	OR (95% C.I.)	OR (95% C.I.)	OR (95% C.I.)
**Model 1**						
**Education attainment**					*fully adjusted model*	
elementary (ref. cat.)	1 *		1 *	1 *	1 *	
secondary	2.12 (1.07–4.21) *		2.36 (1.17–4.75) *	2.16 (1.08–4.30) *	2.42 (1.19–4.90) *	
university	1.11 (0.46–2.64)		1.16 (0.48–2.79)	1.03 (0.43–2.48)	1.08 (0.45–2.62)	
**Gender** (males vs. females)	1.42 (0.83–2.42)	1.52 (0.87–2.63)		1.66 (0.95–2.91)	1.67 (0.95–2.94)	
**Age** (continual)	0.98 (0.96–1.01)	0.98 (0.96–1.00)	0.98 (0.96–0.99) *		0.98 (0.96–0.99) *	
**Model 2**						
**Vaccination**						*fully adjusted model*
none (ref. cat.)	1 *			1 *	1 *	1 *
first dose	1.18 (0.21–6.77)			1.11 (0.19–6.57)	0.93 (0.16–5.52)	0.93 (0.15–5.60)
second dose	1.67 (0.8–3.52)			1.52 (0.71–3.25)	1.51 (0.70–3.23)	1.48 (0.69–3.18)
third dose	0.50 (0.27–0.91) *			0.47 (0.25–0.88) *	0.44 (0.23–0.82) **	0.46 (0.24–0.88) *
**Education attainment**						
elementary (ref. cat.)	1 *			1 *	1 *	1 *
secondary	2.12 (1.07–4.21) *			2.29 (1.14–4.63) *	2.35 (1.16–4.76) *	2.53 (1.23–5.20) *
university	1.11 (0.46–2.64)			1.49 (0.60–3.71)	1.40 (0.56–3.51)	1.43 (0.57–3.60)
**Gender** (males vs. females)	1.42 (0.83–2.42)				1.86 (1.05–3.51) *	1.85 (1.03–3.30) *
**Age** (continual)	0.98 (0.96–1.01)					0.98 (0.96–1.00)

Abbreviations: OR—odds ratio (logistic regression coefficient); C.I.—confidence interval; ref. cat.—reference category; * *p* < 0.05.

**Table 7 viruses-14-01646-t007:** Logistic regression models testing the crude and adjusted effects of four variables on being vaccinated (at least once) against COVID-19, cumulatively adjusted in four models.

	Crude Effect	Model 1	Model 2	Model 3	Model 4
	OR (95% C.I.)	OR (95% C.I.)	OR (95% C.I.)	OR (95% C.I.)	OR (95% C.I.)
**Non-Roma vs. Roma**	3.39 (1.61–7.16) ***	2.86 (1.33–6.13) **	2.87 (1.33–6.19) **	2.80 (1.29–6.10) **	2.97 (1.35–6.55) **
**PsO duration**	1.03 (1.01–1.04) **	1.02 (1.00–1.04) *	1.03 (1.01–1.04) **	1.03 (1.01–1.05) **	1.03 (1.01–1.05) **
**PsO**—**scalp**	0.68 (0.41–1.13)		0.56 (0.32–0.96) *	0.54 (0.31–0.93) *	0.54 (0.31–0.93) *
**Number of comorbidities**	0.88 (0.78–0.99) *			0.86 (0.76–0.97) *	0.86 (0.75–0.97) *
**Gender (women vs. men)**	1.80 (1.12–2.89) *				1.86 (1.13–3.07) *

Abbreviations: OR—odds ratio (logistic regression coefficient); C.I.—confidence interval; PsO—psoriasis; * *p* < 0.05; ** *p* < 0.01; *** *p* < 0.001.

**Table 8 viruses-14-01646-t008:** Prevalence (numbers and relative prevalence in %) of five levels of COVID-19 outcome severity among psoriatic patients stratified according to three groups of provided biological therapy, differences tested with Pearson’s chi-squared test.

Biological Therapy		COVID-19 Outcome					Chi2 Value *p*-Value	Total
		Latent	Mild	Severe	Very Severe	Critical (Death)	*p*-Value	
**anti-TNF**	N	6	12	1	2	0	4.710 *0.788*	21
	%	28.6%	57.1%	4.8%	9.5%	0.0%		100.0%
**anti-IL17**	N	3	12	1	0	0		16
	%	18.8%	75.0%	6.3%	0.0%	0.0%		100.0%
**anti-IL23**	N	5	18	3	3	1		30
	%	16.7%	60.0%	10.0%	10.0%	3.3%		100.0%
**Total**	N	14	42	5	5	1		67
	%	20.9%	62.7%	7.5%	7.5%	1.5%	100.0%	100.0%

Abbreviations: TNF—tumour necrosis factor.

## Data Availability

Data available on request due to privacy.

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
