# Peer review of "Vaccination, Risk Factors and Outcomes of COVID-19 Infection in Patients with Psoriasis—A Single Centre Real-Life Experience from Eastern Slovakia"

_viruses, 2022, doi:10.3390/v14081646_

Round 1

Reviewer 1 Report

The paper is acceptable for publication

Reviewer 2 Report

The manuscript by Baloghová et al. reported an interesting study to identify factors associated with contracting COVID-19 and determining the severity of COVID-19 among psoriatic patients in a real practice setting. They found the protective effect of vaccination, especially the third-dose, against the COVID-19 outcome. The results are interesting and would have significant impact on the COVID-19 research. Therefore, I would recommend its acceptance for publication after minor revision.

1) Plasma concentrations of proinflammatory molecules in patients with SARS‐CoV‐2 were discussed, but no positive conclusions were obtained. I would suggest to summarize there factors with some representative papers.

2)  Factors associated with vaccination status were discussed, however more informations related to the types of vaccination should be added.
